# Increased Homer Activity and NMJ Localization in the Vestibular Lesion *het^−/−^* Mouse *soleus* Muscle

**DOI:** 10.3390/ijms25168577

**Published:** 2024-08-06

**Authors:** Gabor Trautmann, Katharina Block, Martina Gutsmann, Stéphane Besnard, Sandra Furlan, Pierre Denise, Pompeo Volpe, Dieter Blottner, Michele Salanova

**Affiliations:** 1Institute of Integrative Neuroanatomy, Neuromuscular Signaling and System, Charité—Universitätsmedizin Berlin, Corporate Member of Freie Universität Berlin, Humboldt-Universität zu Berlin, and Berlin Institute of Health, Philippstrasse 12, 10115 Berlin, Germany; gabor.trautmann@charite.de (G.T.); katharina.block@charite.de (K.B.); martina.gutsmann@charite.de (M.G.); dieter.blottner@charite.de (D.B.); 2Center of Space Medicine Berlin, 10115 Berlin, Germany; 3UR VERTEX 7480, CHU de Caen, Université de Caen Normandie, 10115 Caen, France; stephane.besnard@unicaen.fr; 4C.N.R. Institute of Neuroscience, 14000 Padova, Italy; sfurlan@mail.bio.unipd.it; 5COMETE U1075, INSERM, CYCERON, CHU de Caen, Normandie Université, Université de Caen Normandie, 10115 Caen, France; pierre.denise@unicaen.fr; 6Department of Biomedical Sciences, University of Padova, 14000 Padova, Italy; pompeo.volpe@unipd.it

**Keywords:** Homer, NMJ, vestibular lesion

## Abstract

We investigated the shuttling of Homer protein isoforms identified in soluble (cytosolic) vs. insoluble (membrane–cytoskeletal) fraction and Homer protein–protein interaction/activation in the deep postural calf *soleus* (*SOL)* and non-postural *gastrocnemius* (*GAS*) muscles of *het^−/−^* mice, i.e., mice with an autosomal recessive variant responsible for a vestibular disorder, in order to further elucidate a) the underlying mechanisms of disrupted vestibular system-derived modulation on skeletal muscle, and b) molecular signaling at respective neuromuscular synapses. Heterozygote mice muscles served as the control (CTR). An increase in Homer cross-linking capacity was present in the *SOL* muscle of *het^−/−^* mice as a compensatory mechanism for the altered vestibule system function. Indeed, in both fractions, different Homer immunoreactive bands were detectable, as were Homer monomers (~43–48 kDa), Homer dimers (~100 kDa), and several other Homer multimer bands (>150 kDA). The *het^−/−^ GAS* particulate fraction showed no Homer dimers vs. *SOL*. The *het^−/−^ SOL* soluble fraction showed a twofold increase (+117%, *p* ≤ 0.0004) in Homer dimers and multimers. Homer monomers were completely absent from the *SOL* independent of the animals studied, suggesting muscle-specific changes in Homer monomer vs. dimer expression in the postural *SOL* vs. the non-postural *GAS* muscles. A morphological assessment showed an increase (+14%, *p* ≤ 0.0001) in slow/type-I myofiber cross-sectional area in the *SOL* of *het^−/−^* vs. CTR mice. Homer subcellular immuno-localization at the neuromuscular junction (NMJ) showed an altered expression in the *SOL* of *het^−/−^*mice, whereas only not-significant changes were found for all Homer isoforms, as judged by RT-qPCR analysis. Thus, muscle-specific changes, myofiber properties, and neuromuscular signaling mechanisms share causal relationships, as highlighted by the variable subcellular Homer isoform expression at the instable NMJs of vestibular lesioned *het^−/−^* mice.

## 1. Introduction

In mammals, locomotion is the result of commands generated by cortical neurons in the central nervous system, translated by the ventral spinal α-motor neurons and executed through contractions of skeletal muscle fibers. Afferent input from somatosensory systems, such as the vestibular system, modulates the original cortical signal, enabling the coordination of movement according to the body posture and balance. The vestibular system integrates information about the body’s position, posture, orientation to the gravitational vector, and contact to the surface of the environment and, thus, helps to coordinate the commands required for skeletal muscle contraction to achieve the desired fluent movement.

The central vestibular system interprets information mainly from the peripheral vestibular organ and integrates additional inputs from the proprioceptive somatosensory and visual systems [1]. Accordingly, nuclei of the central vestibular system project to different neural centers such as the cerebellum, the parietal lobe of the telencephalon, and the spinal cord, modulating their function according to the desired movement [2]. For example, the vestibulospinal tract, originated from the vestibular nuclei in the brainstem, is directly connected to α-motor neurons of the postural extensor skeletal muscles located in the lumbosacral region of the spinal cord [3,4], innervating the hind limb musculature in mice through the lumbosacral spinal ventral system. Thus, the general consensus is that the vestibular system modulates skeletal muscle contraction both indirectly and through direct innervation. Further evidence supporting this hypothesis came from disorders of the vestibular system displaying an altered musculoskeletal system. For instance, alteration in the skeletal muscle physiology and excessive myofiber remodeling have been observed in rodents after both chemical and mechanical labyrinthectomy [5,6,7]. In addition, changes in the electromyographic activity of different skeletal muscles (*soleus*, *erector spinae*, and *sternocleidomastoideus*) have been detected in both healthy humans after direct stimulation of the vestibular system and patients with peripheral vestibular lesion [8,9]. Nevertheless, it was previously demonstrated that not only skeletal muscle, but also bone remodeling and homeostasis, are affected by bilateral vestibular lesions in rats, suggesting a diverse influence of the vestibular system on the musculoskeletal system [10]. The modulatory effect of the vestibular system on skeletal muscle, bone structure, and vegetative functions also plays an important role during and after chronic exposure to a microgravitational (μg) environment, such as spaceflight, during which the acceleration sensation is mainly affected [11,12,13,14]. These results clearly suggest a modulatory effect of the vestibular system on skeletal muscle physiology; however, alterations at the cellular and molecular levels are yet not fully understood.

The neuromuscular junction (NMJ) is a unique chemical synapse between the terminal axon of a ventral spinal motor neuron and a skeletal muscle fiber in the periphery. It is responsible for the translation between the nervous and the musculoskeletal systems. The postsynaptic region of the skeletal muscle fiber is adapted to the functionality of the NMJ, maintaining a dense, post-synaptic protein network with many scaffold proteins as the backbone structure [15]. These scaffold proteins play an important role in effectively organizing cellular signaling pathways and coordinating molecular players in different cellular functions in space and time [16]. The Homer protein family is a group of scaffold adapter proteins detected originally at the post-synaptic density of dendrites in hippocampal neurons, but present also in skeletal muscle fibers [17,18,19]. Their interacting partners in skeletal muscles include inositol-3-phosphate receptor type 1 (IP3R1) [20,21], ryanodine-receptor type 1 [22], TRP-Channels [23], and NFATc1 [6], suggesting an important role of the Homer proteins in intracellular Ca^2+^-signaling [24,25]. There are many Homer isoforms due to alternative splicing [19,26]. Long Homer isoforms contain a C-terminal coil-coiled domain, allowing a dynamic homo-and hetero-oligomerization [27] and potential cross-linking of interacting partners [28,29,30]. In skeletal muscle fibers, Homer proteins localize at the Z-disc and at the NMJ, in close proximity to proteins belonging to the intracellular Ca^2+^ homeostasis and signaling pathways [19]. Homer protein expression is suggested to be linked to the motor neuronal activity. For example, Bortoloso et. al. [31] hypothesized that Homer proteins might play a central role in denervation-related muscle atrophy in slow-twitch muscle fibers. Homer protein expression is also downregulated after long-term disuse in both humans and mice [31,32]. In contrast, plantar vibration stimulation along with resistive exercise countermeasures in bed rest showed an increase of the Homer signal at the postsynaptic domain [31].

Thus, we hypothesize that alteration in the peripheral vestibular system and the resulting impaired neuronal activity altering skeletal muscle physiology, and affecting subcellular architecture, could be related to Homer protein expression and regulation as the result of an altered Homer cell signaling. The head tilt knockout (*het^−^/^−^*) mouse lends itself as a good model to investigate specific pathologies of the peripheral vestibular organ, and to study ensuing effects on skeletal muscle structure and function [33]. The *het^−/−^* mouse is devoid of otoconia, calcium-carbonate crystals on the otolithic membrane in the saccule and utricle. Thus, the *het* mutation specifically alters the acceleration and, as such, gravitation sensation [34].

Because several lines of evidence show that Homer signaling is mostly important for slow-twitch skeletal muscles [35], the aim of the present study was to (i) analyze myofiber morphology and phenotype composition in the postural hind limb deep calf soleus (*SOL*, mostly represented by oxidative or slow type-I myofibers) and the non-postural gastrocnemius (*GAS*, mostly represented by glycolytic or fast type-II myofibers) skeletal muscle, and (ii) to investigate alteration in the Homer protein expression and subcellular distribution between the *het^−/−^* variant and heterozygous (CTR) mouse; this might, in turn, be the result of a peculiar subcellular reorganization and altered signaling at the NMJ.

## 2. Results

### 2.1. The het^−/−^ Mouse Is Characterized by Increased Homer Dimer in the Cytosolic Fraction of SOL Muscle and Loss of Both Homer Monomer and Dimer in the Insoluble/Pellet Fraction of GAS Muscle

To further study Homer cell signaling in the skeletal muscle of *het^−/−^* mice, two different types of muscle were investigated: the postural (slow-type myofiber-enriched *SOL*) and the non-postural (fast-type myofiber-enriched *GAS*) muscles. The *SOL* and *GAS* muscles of heterozygote mice were used as the reference control.

Both slow- and fast-twitch *SOL* and *GAS* muscles of *het^−^/^−^* and CTR mice were homogenized in an isotonic buffer in native experimental conditions, in the absence of reducing reagents, and soluble (cytosol) and insoluble (pellet) fractions were obtained by differential centrifugation according to a previously established experimental protocol.

Furthermore, the same amount of proteins in each fraction were separated on a continuous gradient SDS-PAGE gel and investigated by Western blot analysis for the presence of Homer monomer, dimer, and multimeric proteins using specific anti-Homer antibodies [31]. As shown in Figure 1a,c, several anti-Homer antibody immunoreactive bands were detected at the levels of approximately 43–45, 90, and 180 kDa molecular weight, corresponding to the Homer monomers, dimers, and tetramers, respectively. Due to our biochemical experimental protocol, large Homer macromolecular complexes above Homer dimers such as tetramers and/or multimers were not taken into consideration in the present study. In *SOL*, no Homer monomer was detected in both the soluble and insoluble fractions of both *het^−/−^* and CTR mice, suggesting that in the slow-type postural muscle, all Homer proteins were in a dimer or multimer configuration (Figure 1a). Densitometry analysis of the Homer dimer soluble vs. insoluble fraction showed a 4.3 ratio in the *het^−/−^* mouse and a 2.2 ratio in the CTR mouse groups (CTR soluble 436.6 ± 83.55 SEM vs. insoluble 198.4 ± 39.57 SEM; *het^−/−^* soluble 948.3 ± 52.95 SEM vs. insoluble 221.3 ± 52.69 SEM) (Figure 1b). The densitometric analysis showed the Homer dimer soluble fraction of the *het^−^/^−^* mouse was significantly higher than that of the CTR mouse (+117%; CTR soluble 436.6 ± 83.55 SEM vs. *het^−/−^* soluble 948.3 ± 52.95 SEM; *p* ≤ 0.0004); however, no major difference was observed between the insoluble fraction of the *het^−/−^* and the insoluble fraction of the CTR mice group (CTR insoluble 198.4 ± 39.57 SEM vs. *het^−/−^* insoluble 221.3 ± 52.69 SEM; *p* ≤ 0.74) (Figure 1b).

In the *GAS* of CTR mice, all Homer monomer, dimer, and tetramer/multimer configurations were present in both the soluble and insoluble fractions (Figure 1c). Comparing the soluble vs. insoluble fractions of Homer proteins in the *GAS* muscle showed a 1.45 ratio in the *het^−/−^* mouse and a 20.5 ratio in the CTR mouse group (CTR soluble 443.6 ± 136.9 SEM vs. insoluble 21.66 ± 10.41 SEM; *het^−/−^* soluble 503.8 ± 110 SEM vs. insoluble 205.1 ± 34.29 SEM) (Figure 1d).

### 2.2. Increased Homer and Nicotinic Acetylcholine Receptor (nAChR) Protein Expression at the NMJ Postsynaptic Microdomain of het^−/−^ Mouse SOL

Immunohistochemistry experiments were performed in order to further investigate the Homer protein expression and subcellular compartmentalization at the NMJ postsynaptic microdomain of both *SOL* and *GAS* muscles of *het^−/−^* and CTR mice and to correlate them to the nAChR clustering. As reported in Figure 2a,c, an increase in Homer immunosignals (+40%), expressed as fluorescence intensity, in the *SOL* NMJ postsynaptic microdomain of the *het^−/−^* mouse was observed when compared to the CTR *SOL* (CTR 70.98 ± 9.338 SEM, *n* = 53, vs. *het^−/−^* 100.2 ± 9.159 SEM, *n* = 69, *p* = 0.0497) (Figure 2a). In contrast, no significant fluorescence intensity changes were present in the NMJ of the *GAS* muscle between the *het^−/−^* and CTR groups (CTR 91.72 ± 5.058 SEM, *n* = 71, vs. *het^−/−^* 91.11 ± 9.822 SEM, *n* = 45, *p* = 0.9571) (Figure 2c).

Figure 2b,d shows a representative NMJ image of *SOL* (Figure 2b) and *GAS* (Figure 2d) muscles of CTR and *het^−/−^* mice immunostained with anti-pan-Homer antibodies.

Since the *het^−/−^* mice have an increased daily activity when compared to an age- and sex-matched cohort of CTR mice, we sought to investigate nAChRs density at the postsynaptic microdomain as potential NMJ structural changes between the two types of muscle and animal groups. To this aim, we used a conventional fluorophore-labeled α-bungarotoxin (BTx) known to specifically bind to nicotinic acetylcholine receptors. As reported in Figure 3, both the *SOL* (Figure 3a) and the *GAS* (Figure 3b) muscles of the *het^−/−^* mouse group were characterized by an increased mean BTx signal intensity at the NMJ (*SOL*: CTR 36.61 ± 1.527 SEM, *n* = 52, vs. *het^−/−^* 49.33 ± 1.645 *SEM*, *n* = 177, *p* = 0.0013; *GAS*: CTR 42.52 ± 1.115 SEM, *n*= 77, vs. *het^−/−^* 49.17 ± 1.192 SEM, *n* = 124, *p* = 0.0002).

Thus, taken together, these results are suggestive of an increase in Homer and nAChR signaling at the NMJ postsynaptic microdomain of the *het^−/−^ SOL* muscle.

### 2.3. Homer 1b/c in the GAS and H2a/b in the SOL Muscles Are the Dominating Expressed Homer Isoforms

To further characterize the expression pattern of the Homer protein family in the skeletal muscle of *het^−/−^* vs. heterozygous CTR mice, the presence of specific Homer transcripts for the short form of Homer 1a, and the long forms of Homer 1b/c and Homer 2a/b were investigated by qPCR experiments.

As shown in Figure 4a, in the *SOL* muscle, the relative expression level of Homer 2a/b transcript was the highest (CTR 3.28 ± 0.3525 SEM vs. *het^−/−^* 3.7 ± 0.2071 SEM, *p* = 0.3333), followed by Homer 1b/c (CTR 1.25 ± 0.0265 SEM vs. *het^−/−^* 1.34 ± 0.0633 SEM, *p* = 0.1797), and finally, Homer 1a (CTR 0.266 ± 0.0251 vs. *het^−/−^* 0.225 ± 0.0725 SEM, *p* = 0.1898) (Figure 4a).

In contrast, as shown in Figure 4b, in the *GAS* muscle, the highest expression level relative to the reference gene was Homer 1b/c transcript (CTR 2.35 ± 0.2837 SEM vs. *het^−/−^* 2.41 ± 0.1399 SEM, *p* = 0.6993), followed by the Homer 1a (CTR 0.636 ± 0.1191 SEM vs. *het^−/−^* 0.543 ± 0.0473 SEM, *p* = 0.4960), and finally, Homer 2a/b isoform (CTR 0.3 ± 0.0416 SEM vs. *het^−/−^* 0.37 ± 0.03 SEM, *p* = 0.2210) (Figure 4b).

As shown in Figure 4a,b, the qPCR analysis did not reveal significant differences for the transcripts investigated in both the *SOL* and *GAS* muscles between *het^−/−^* and CTR mice. However slow- and fast-type muscles express different Homer isoforms as dominant forms. Our qPCR data align with those of previous studies, showing that Homer 2a/b is the dominant expressed Homer isoform in the SOL muscle [35], whereas Homer 1b/c is the dominant isoform expressed in the GAS muscle, which either lacks the Homer 2a/b isoform or expresses it at a very low level [35].

### 2.4. The het^−/−^ Mouse Is Characterized by a Higher Myofiber Cross-Sectional Area in the Postural SOL Muscle

In order to further investigate the structural and metabolic adaptation of skeletal muscles to the absence of the *het* gene, myofiber cross-sectional area and phenotype composition analysis was performed in both the *SOL* and *GAS* muscles of *het^−/−^* vs. CTR mice. Immunohistochemistry experiments using slow- and fast-type myosin heavy chain (MyHC) antibodies were performed in combination with anti-Dystrophin antibodies, the latter to specifically mark the outer myofiber cell membrane.

As shown in Figure 5a,b the *SOL* muscle of the *het^−/−^* mouse was characterized by a higher mean CSA of both type I (slow-twitch) and type II (fast-twitch) myofibers (CTR type I: 2.220 ± 44.46 μm^2^, *n* = 306, type II 1.954 ± 30.49 μm^2^, *n* = 545, vs. *het^−/−^* type I: 2540 ± 46.13 μm^2^, *n* = 542; type II: 2234 ± 33.91 μm^2^, *n* = 775, *p* ≤ 0.0001) and by a non-significant phenotype shifting towards the type II myofibers (CTR type I: 64.8% type II: 34.2% vs. *het^−/−^* type I: 60.5% type II: 38.2%) (Figure 5b–d). A significant difference in the CSA and myofiber phenotype composition was not observed in the *GAS* muscle (CTR type II: 2738 ± 47.07 μm^2^, *n* = 2666, vs. *het^−/−^* type II: 2666 ± 49.21 μm^2^, *n* = 2738, *p* = 0.2877) (Figure 5a).

## 3. Discussion

In the present study, by using an experimental animal model of the *het^−/−^* mouse, we gathered experimental evidence suggestive of a potential role of the vestibular system in the regulation of skeletal muscle structure and function. Our data indicate that in the absence of an efficient peripheral vestibular system, altered myofiber type-I and type-II cross-sectional area can be observed along with altered postsynaptic nAChR density and Homer protein expression in the postural *SOL* muscle only.

Thus, the *het^−/−^* mouse is a well-accepted lesion model and an excellent candidate to investigate the modulatory effect of a functional vs. non-functional peripheral vestibular system, specifically, the acceleration and, thus, the gravitation sensation on the entire body and on the skeletal muscle structure. The effect of the *het* mutation present in mice mimics, at least in part, some of the vestibular restrain previously observed in astronauts in spaceflight, notably, that weightlessness predominantly affects the saccule and utricle, but not the semicircular, channels [36].

Quite interestingly, the *het^−/−^* mouse shows the characteristics observed in mice with peripheral vestibular lesion, including abnormal circling behavior, flexion towards the abdomen if hanged by the tail, and a lack of ability to swim properly [33,37]. However, although this model itself is considered promising to understand a specific aspect (saccule/utricle) of the connection between the peripheral vestibular system and skeletal muscle physiology, because of the congenital nature of the “lesion”, further investigation is needed to better understand a possible structural anatomical adaptation of the central nervous system to the altered peripheral vestibular input.

Notably, the newborn *het^−/−^* mouse grows up without peripheral vestibular acceleration sensation, which may lead to altered neuronal development and adaptation mechanisms in other organ systems as well [11]. Other peripheral sensory information besides the acceleration sensation plays an important role in whole body balance and locomotion, such as proprioceptors in the joints [38] and visual input [39], which may overtake a more central function in conditions without an adequate input from the peripheral vestibular organ of the inner ear.

Based on the results of our experiments, morphological alterations were present between the postural and non-postural muscles of *het^−/−^* vs. heterozygote CTR mice. Indeed, the morphological analysis revealed that the *het^−/−^* mutation predominantly affected the myofiber cross-sectional area in the *SOL* muscle but not in the *GAS* muscle. Surprisingly, we observed a myofiber phenotype shifting in the *SOL* muscle towards the fast myofiber phenotype (type I > II shift). These results correlate well with the concept that although the *SOL* and *GAS* calf muscles share a location in the lower limb (calf triceps), they fulfill a different functionality during locomotion and, thus, show different myofiber composition accordingly [5,6]. The soleus muscle is responsible for energy efficiency, endurance, and/or long-duration locomotion and adjusting postural balance, whereas the gastrocnemius muscle, with its more than 90% type II myofiber composition, is considered as the “executer” for fast, explosive movements during active walking, running, or jumping.

Thus, our data suggest that a congenital lesion affecting the vestibular system in *het^−/−^* mice has a much stronger impact on the soleus muscle physiology than on the gastrocnemius muscle, with an obvious impact on postural control.

Moreover, our results showed an increased NMJ reorganization of the postural soleus vs. the non-postural gastrocnemius muscle in *het^−/−^* mice with vestibular lesion. The higher nAChR-signal intensity and the altered Homer expression and distribution, especially the increased concentration at the NMJ, suggest that the *het^−/−^* mouse is characterized by a different neuronal activity affecting both skeletal muscle myofiber structure and postsynaptic molecular architecture. In fact, the increase in the Homer 2 isoform mRNA concentration is present only in type I myofibers and is located mainly at the postsynaptic region of the NMJ [35]. Interestingly, previous studies showed that a higher frequency in the neuronal action potential rate results in an increased nAChR-subunit [40] and Homer protein expression at the NMJ [27], and that the Homer protein expression is increased after exercise [31], raising the possibility that the *het^−/−^* mouse soleus muscle receives a yet-unknown diversity of neural inputs from spinal motor neurons. Thus, a higher neuronal input could explain the increased cross-sectional area found in the *het^−/−^* mouse, with an adaptation of the corresponding NMJ postsynaptic microdomain. Furthermore, our data point towards a more prominent alteration in type I muscle fibers compared to type II fibers, as judged by the increased myofiber CSA, higher Homer 2 isoform presence, and focused NMJ reorganization in the soleus muscle as compared to the gastrocnemius muscle, representing functionally distinct parts of the calf triceps surae muscle involved in posture and locomotion. Interestingly, we found an almost twofold higher Homer dimer signal in the soluble (cytosolic) fraction of the *het^−/−^* mouse without any difference in the insoluble (membrane-cytoskeletal) fraction. Since only a slight, non-significant increase of Homer long isoform mRNA expression has been observed, either an increased protein translation or a decreased degradation of these proteins could explain the higher protein pool, ready to oligomerize upon the neural input. Indeed, a dynamic molecular assembly upon neuronal activity has previously been demonstrated in the central nervous system; however, until now, this dynamic has not been investigated in skeletal muscle myofibers [27]. The higher Homer signal intensity at the postsynaptic NMJ in the *het^−/−^* mouse could not explain, solely, a high Homer concentration in the cytosolic fraction, raising the possibility of an increased molecular accumulation of Homer signals close to the Z-discs, representing the mechanical links between actin–actin myofibrils of the subcellular myofiber contractile apparatus.

Based on the tissue fractionation protocol used in this study, the cytosolic fraction contains proteins that are not or not-tightly bound to membrane or cytoskeletal cell components opposite to the membrane–cytoskeletal fraction [41,42]. An increased cytosolic Homer signal ratio vs. the membrane–cytoskeleton fraction and Homer dimerization found in our study supports our notion for a higher overall Homer protein expression and cross-linking capacity.

Although myofiber CSA was not significantly affected in the gastrocnemius muscle, altered peripheral vestibular input changed the overall Homer protein expression (overall decrease except Homer 1a isoform) and, surprisingly, an inverse nAChR concentration at the NMJ was observed. In addition, the gastrocnemius of the *het^−/−^* mouse did not show any difference in Homer isoform transcriptome expression, yet the overall Homer protein concentration was decreased, suggesting either a limited protein translation or an increased rate of protein degradation compared to the heterozygote mouse. Homer protein signal intensity at the postsynaptic NMJ showed no detectable difference; therefore, lower protein concentrations at the Z-disc myofibrillar microdomains are to be expected.

As mentioned above, our data point towards the possibility that the *het^−/−^* mouse is characterized by a higher neuronal input of the postural skeletal muscle soleus. On the other hand, the gastrocnemius muscle showed an overall lower Homer protein concentration without changes in the CSA. Although the *het^−/−^* mouse indeed shows hyperactive behavior [33], this hyperactivity might not be the only explanation for a possible higher neuronal input. A shifted neuronal activity profile towards the soleus muscle after alteration in the peripheral vestibular input could also be the cause of an overall higher neuronal input for muscle tone regulation and balance, as reported previously [5].

Different experimental models of disuse-induced muscle atrophy and exercise countermeasure in human and animal models of denervation showed that both Homer expression and subcellular localization at the NMJ are dependent on muscle and nerve activity [31]. Therefore, we assume that the increased Homer expression and the cross-linking activity in the soleus muscle of *het^−/−^* mice, as reported in the present study, are due to an increased muscle and nerve activity caused by altered vestibular input at the NMJ. This leads, in turn, to an increase in NMJ signaling, which may require an increased recruitment of signaling molecules and, thus, cross-linking activity of Homer-dependent signaling pathways.

Quite interestingly, a recent study showed that the NMJ is closely connected to sympathetic axon terminals possibly involved in the modulation of synaptic maintenance [43]. However, to investigate directly the effect of accurate neural activity on the innervating peripheral nerves with the appropriate methodology in the *het^−/−^* mouse model, further functional analysis, for example, by in vivo electromyography, is required.

## 4. Conclusions

Our novel findings in both the *het*-mice *GAS* and *SOL* muscles support previous observations that the peripheral vestibular system affects skeletal muscle structure and function and these mechanisms involve, partially, the Homer protein signaling pathways, playing an important role in Ca^2+^ homeostasis and myofiber atrophy [31,35]. However, further studies are necessary to assess the importance of these effects on skeletal muscle atrophy in patients with vestibular lesions, or even in astronauts, who experience a high-degree of microgravity-induced muscle atrophy that may be at least partially also attributed to an altered vestibular input in vivo following whole body unloading in space.

## 5. Study Limitations

This study is exclusively experimental and has, so far, only been carried out on the muscles of murine models. An equally thorough experimental analysis on the muscles of patients with associated balance and locomotion dysfunction is of fundamental importance and necessary in order to be able to draw conclusions pertaining to human pathology.

## 6. Materials and Methods

### 6.1. het^−/−^ Mice

One group of heterozygote mouse (CTR male, *n* = 6) and one group with the *het^−/−^* mutation (*het^−/−^* male, *n* = 6) were used in this study. The *het^−/−^* mice were purchased at the Charles River Laboratory. The animals were kept in an animal facility at the temperature 21–22 °C in a 330 cm^2^ sized cage, with supplies ad libitum, and a 12–12 h day–light cycle (8 a.m.–8 p.m.). At an age around 15 weeks, the animals were euthanized with isoflurane-anesthesia according to the animal rights guidelines at the University of Caen. Directly after the mice were euthanized, the *SOL* and *GAS* calf muscles were dissected and snap frozen in liquid nitrogen and kept at −80 °C until further procedure.

### 6.2. RNA Extraction and Quantitative Polymerase Chain Reaction (qPCR) Analysis

Total RNA was isolated from mouse *SOL* and *GAS* muscles of each experimental group (homozygous ^−/−^ *n* = 9, heterozygous ^+/−^ *n* = 5) using TRIzol^®^ (Thermo Fisher Scientific, USA). RNA was converted to cDNA by using random hexamers and SuperScript^®^ VILO™ (Thermo Fisher Scientific, Carlsbad, CA, USA). Specific primers for *Homer1a* and *Homer1b/c*, *Homer2*, PPIA, TBP, and HPRT1 were as previously reported [44]. qPCR was performed in a CFX Connect^®^ Real time System (Bio-Rad, Hercules, CA, USA) using SYBR Green chemistry. All samples were run in triplicate, in parallel for each individual muscle sample and simultaneously with RNA-negative controls. *Cyclophilin A* (PPIA), *TATA-Box Binding Protein* (TBP), and *Hypoxanthine Phosphoribosyltransferase 1* (HPRT1) were tested as candidate reference genes. HPRT1 was used to normalize Ct values by ΔCt method. Data are expressed as means +/− SEM. Comparisons were made by using unpaired *t*-test, with *p ≤* 0.05 being considered statistically significant.

### 6.3. Myofiber Cross-Sectional Area and Phenotype Composition Analysis

Frozen muscle tissue was embedded in tissue freezing medium (TissueFreezingMedium, 14020108926, Leica Biosystems, Nussloch, Germany) and sliced to 10 μm thick transversal cryosections at the middle portion of the muscle at −25 °C using a cryostat (Kryostat Frigocut 2800E, Reichert-Jung, Germany). The cryosections thaw-mounted slides were stored frozen at −80 °C, and 2 h prior to immunohistochemical staining, stored at −20 °C in a freezer. To assess the alteration in myofiber CSA and phenotype composition, immunohistochemical analysis was performed using primary antibodies and fluorochrome-labeled secondary antibodies. The sections were fixed with 4% phosphate buffered-formaldehyde (Histofix Carl Roth, Karlsruhe, Germany) for 10 min, washed in TRIS-buffered saline, and inserted in an isolated, humid chamber. Non-specific binding sites were neutralized for 45 min at 21 °C room temperature using 10% “Mouse on Mouse” (M.O.M) blocking reagent in blocking buffer (TRIS; 2% goat serum; 0.3% Triton X) prior to the incubation with the primary antibody. Primary antibodies were used against slow MyHC (mAb, Sigma M8421, 1:2000, Saint Louis, MO, USA), fast MyHC (mAb, Sigma M4276, 1:2000) and dystrophin (pAb, Santa Cruz SC15376, 1:1000, Santa Cruz, CA, USA) and were detected with fluorochrome labeled secondary antibodies, Alexa Fluor at 488 nm (Molecular Probes A11029, 1:3000, Eugene, OR, USA), 555 nm (Invitrogen A21424, 1:3000, Waltham, MA, USA) and 635 nm (Molecular Probes A31577, 1:1000) wavelength respectively. The sections were mounted in DAPI mounting medium. The staining protocol included negative controls lacking the incubation with primary or secondary antibodies. Images were taken with a confocal laser scanning microscope (Leica TCS Sp-8 with Leica confocal software version 2.61 build 1537; Leica Microsystems Heidelberg GmbH, Heidelberg, Germany). The images have been acquired using the same image settings during acquisition. Cross-sectional area and phenotype composition were measured using SMASH MatLab Software 1.0 [45].

### 6.4. Tissue Fractionation and Western Blot Analysis

Each sample was homogenized in a Lysis Buffer 1° (250 mM Succrose, 20 mM Tris-HCL, 1 mM EDTA, pH 7.4 + Protease Inhibitors (cOmplete Tablet EASYpack, Roche, LOT 41353800)) 10 μL/mg tissue ratio with a glass homogenizer (20 times) on ice. After 30 min incubation in the Lysis Buffer 1°, samples were centrifuged with 14.000 g for 20 min at 4 °C. The supernatant was carefully collected and was stored at −80 °C, used later as the soluble (cytosolic) fraction. The same protocol was repeated with the pellet, using Lysis Buffer 2° (1% Triton X-100, 150 mM NaCl, 50 mM TRIS-HCL, 1% (*w*/*v*) NP-40, 0.1% (*w*/*v*) SDS, 1% (*w*/*v*) Na-Deoxycholate + protease inhibitors) 5 μL/mg tissue ratio for homogenization and centrifugation with 14.000 g for 20 min. This supernatant was carefully collected and was stored at −80 °C, used later as the unsoluble (membrane-cytoskeletal) fraction. After measuring the sample protein concentration (Protein Assay Kit, Thermo Scientific, Carlsbad, CA, USA), a portion of the stock solutions was diluted to the same protein concentration (1 mg/mL) with Laemmli sample buffer (0.5 M TRIS, 50% (*v*/*v*) Glycerol, 4% (*w*/*v*) SDS and 0.02% (*w*/*v*) Bromphenolblue) with or without reducing agents (700 mM DDT, 10% β-Mercaptoethanol). Reducing conditions included an additional step of heating to 95 °C for 10 min.

The samples were loaded to a 4–15% gradient-gel (Mini-Protean TGX, BioRad#456-1083, Feldkirchen, Germany) and SDS-PAGE was performed. The gel was blotted on a nitrocellulose membrane (0.45 μm thickness) at 100 V for 60 min. After blocking with 2% (*w*/*v*) slim milk powder (SERVA, 425,590.01, LOT 200120, Hidelberg, Germany) at room temperature for 60 min, we used affinity-purified primary antibodies (Salanova, 1:250, 60 min, RT.) to detect the Homer proteins. Alkaline-phosphatase conjugated secondary antibodies (DAKO D0306, 1:500, 60 min, RT) were used with an alkaline-phosphatase based detection system (NBT/BCIP, #34042, Thermo Scientific, USA) to detect the primary antibodies. Mouse brain lysate was used as a positive control, because neural tissue contains Homer proteins in high concentrations [19].

### 6.5. Immunohistochemistry

To determine the difference in the in vivo distribution of the Homer proteins in skeletal muscle between the two groups, immunohistochemical analysis was carried out with the same primary antibody we used for the Western blot analysis and secondary antibody labelled with fluorophores (Alexa 488 nm, Molecular Probes A11034, 1:1000). Acetylcholine receptors were highlighted with fluorophore-conjugated alpha-Bungarotoxin (555 nm, Invitrogen B35451, 1:1000). Images were taken with the same confocal imaging system used for the CSA analysis (see above). The Homer signal intensity was measured at the post-synaptic region of the neuromuscular junction. The post-synaptic region was determined as the area from the BTx-labelled nAChRs to a maximum of 5 μm deep border.

### 6.6. Statistical Analysis

Statistical analysis was carried out by Graph-Pad^®^ PRISM software 9.5.1. Data were subjected first to descriptive statistics and to multiple normality tests (D’Agostino and Pearson test, Anderson–Darling test, Shapiro–Wilk test, and Kolmogorov–Smirnov test) and the method of analysis was decided according to the distribution of the data sets. Data with normal Gaussian distribution were analyzed by unpaired, two-tailed *t*-test with Welch’s correction and data with non-Gaussian distribution were analyzed using Mann–Whitney test. Data are expressed as means +/− SEM. Results of analysis were considered statistically significant at a predefined level of *p* ≤ *0.05*. The number of samples analyzed are indicated as *n* = X at the corresponding result and/or graphic section.

## Figures and Tables

**Figure 1 ijms-25-08577-f001:**
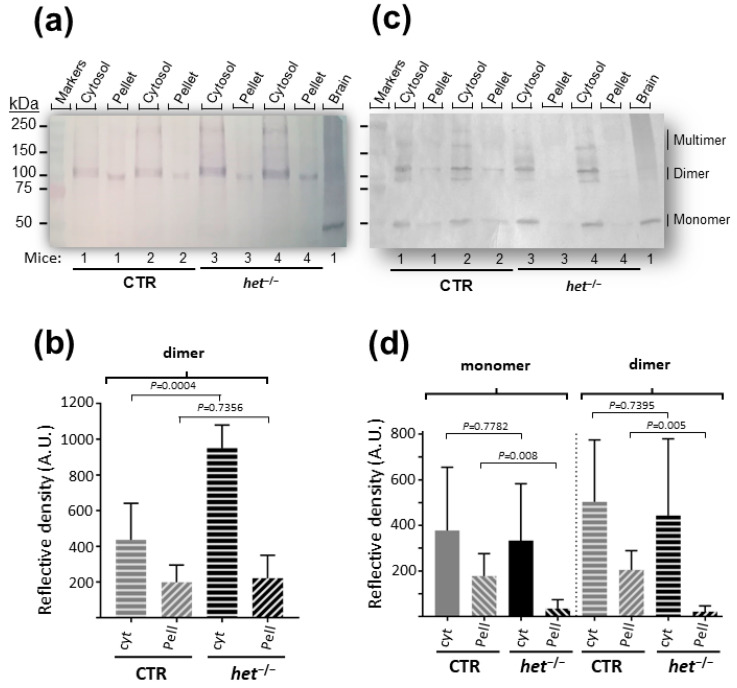
Homer skeletal muscle Western blot analysis from *SOL* (**a**,**b**) and *GAS* (**c**,**d**) muscle. (**a**) Representative image of soluble (cytosol) and insoluble (pellet) fractions of CTR and *het^−/−^* mice. Homer dimer (~90 kDa) and Homer multimer (above 150 kDa) immunoreactive bands are present in both soluble and insoluble fractions of both animal groups. (**b**) Homer quantification by densitometry analysis in *SOL*. (**c**) Representative image of soluble (cytosol) and insoluble (pellet) fractions of CTR and *het^−/−^* mice. Homer monomer (~43–45 kDa), dimer (~90 kDa), and multimer (above 150 kDa) immunoreactive bands are detectable mostly in the soluble fraction. No Homer immunoreactive bands are present in the insoluble fraction of *het^−/−^* mice. (**d**) Homer quantification by densitometry analysis in *GAS*. Cytosol = soluble fraction; pellet = insoluble fraction. Brain was used as positive internal control in panels a and c. Results are from at least three independent experiments. Data were compared using unpaired Student’s *t*-test and considered statistically significant at *p* ≤ 0.05. Altogether, a total of *n* = 6 CTR and *n* = 6 *het*^−/−^ mice were investigated.

**Figure 2 ijms-25-08577-f002:**
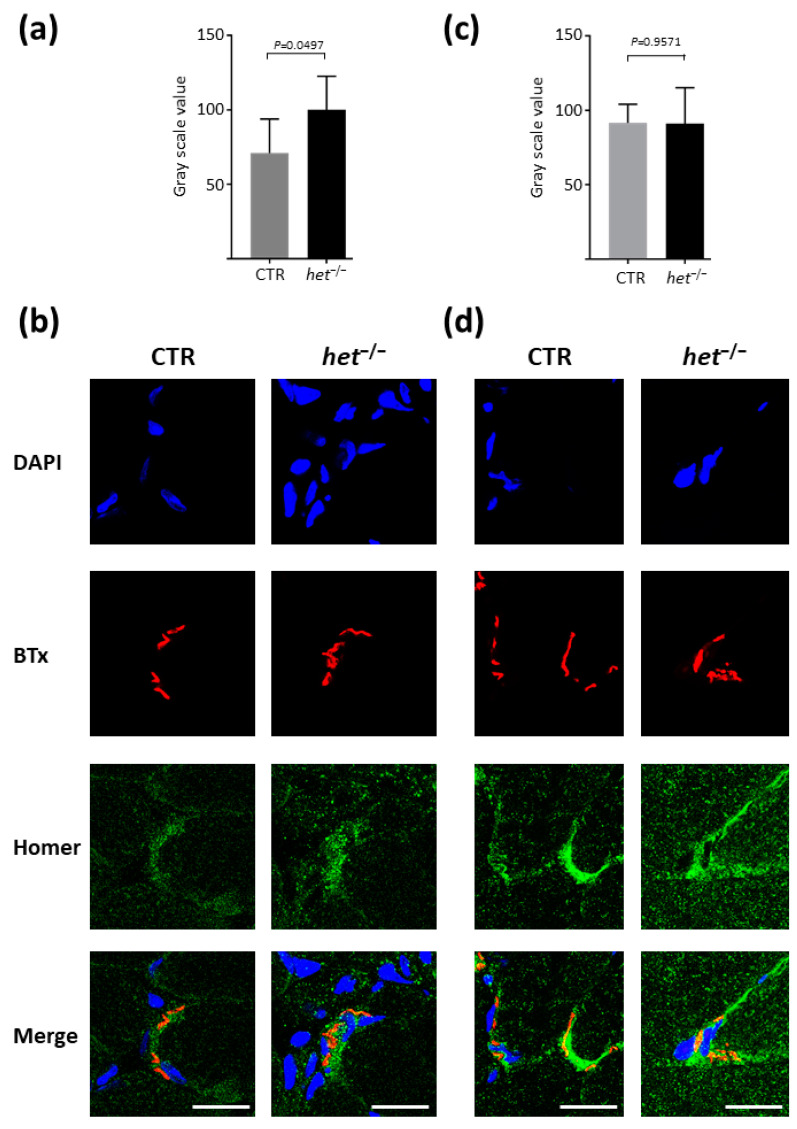
Homer immunofluorescence intensity at the NMJ. (**a**,**b**) *SOL* muscle. (**a**) The graphs show the mean gray scale value obtained in *SOL* from the Homer signal at the NMJ postsynaptic region. (**b**) Representative Homer immunofluorescence analysis in CTR and *het^−/−^* of NMJ *SOL* muscle. The number of analyzed NMJs in the *SOL*: *het^−/−^ n* = 69; CTR *n* = 52. (**c**,**d**) *GAS* muscle. (**c**) The graphs show the mean gray scale value obtained in *GAS* from the Homer signal at the NMJ postsynaptic region. (**d**) Representative Homer immunofluorescence analysis in CTR and *het^−/−^* of NMJ *GAS* muscle. The number of analyzed NMJs in the *GAS*: *het^−/−^ n* = 45; CTR *n* = 71. In blue = DAPI cell nuclei staining; red = α-Bungarotoxin (BTx)-labelled AChRs; green = Homer proteins. Magnification scale bar = 15 μm. Data were compared using unpaired Student’s *t*-test and/or Mann–Whitney test and considered statistic significant at *p* ≤ 0.05.

**Figure 3 ijms-25-08577-f003:**
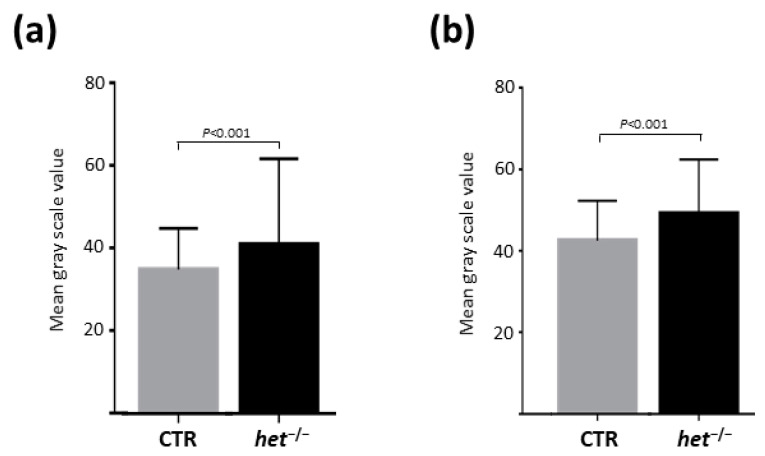
nAChRs fluorescence intensity analysis at the NMJ. (**a**) *SOL* muscle; (**b**) *GAS* muscle. The graphic shows the mean value of the fluorophore-conjugated BTx signal, marking the nAChRs at the postsynaptic domain of the NMJ of CTR (gray column) and *het^−/−^* (black column). The number of NMJs analyzed: SOL *het^−/−^ n* = 177; CTR *n* = 52; GAS *het^−/−^ n* = 124; CTR *n* = 77.

**Figure 4 ijms-25-08577-f004:**
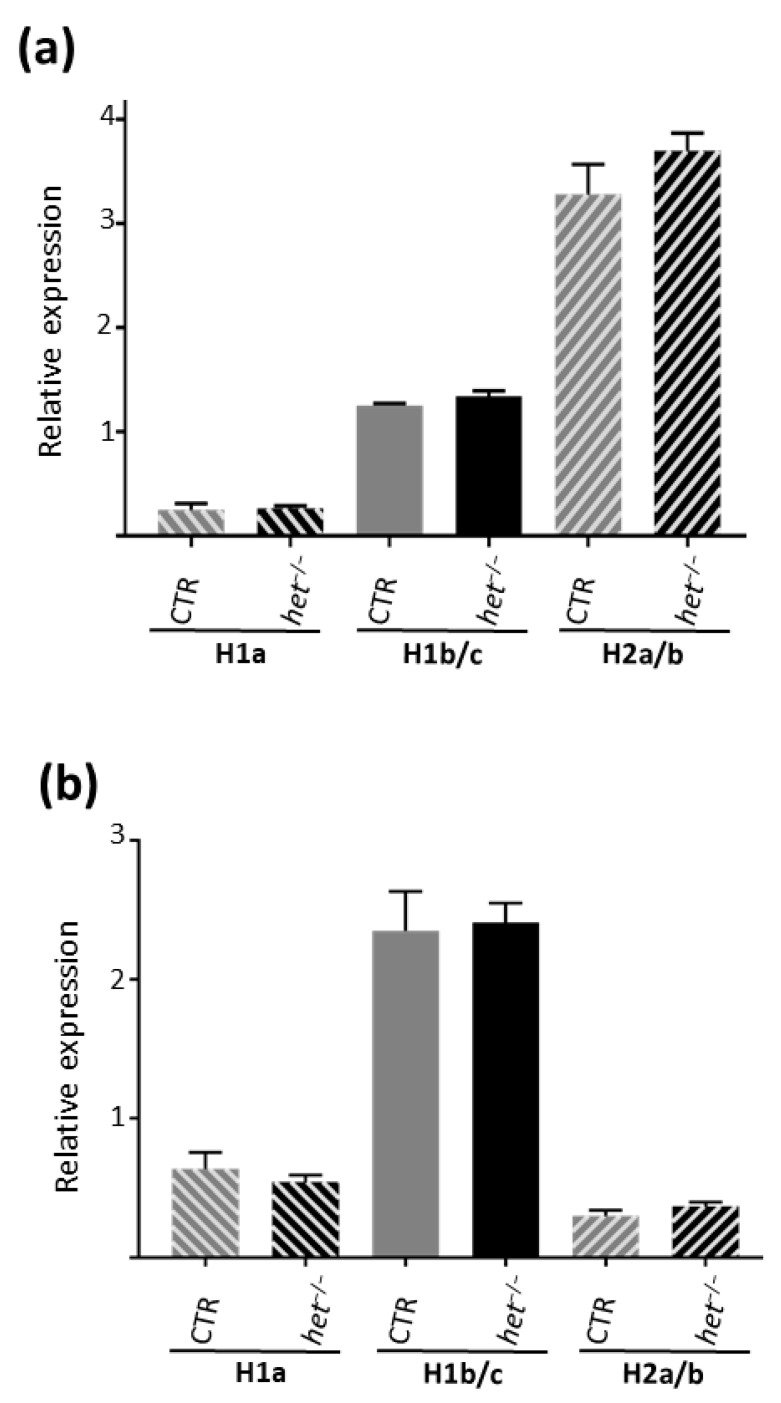
Homer specific isoforms transcripts expression analysis by qPCR. (**a**) *SOL* muscle; (**b**) *GAS* muscle. Graphs represent the relative expression of Homer-isoform-specific mRNA to the housekeeping gene HPRT1 in the *SOL* and *GAS* muscles, respectively. In the *GAS* muscle, the dominating Homer isoform is the long Homer 1b/c, whereas in the *SOL*, it is Homer 2a/b. H1a = Homer 1a; H1b/c = Homer 1b/c; H2a/b = Homer 2a/b.

**Figure 5 ijms-25-08577-f005:**
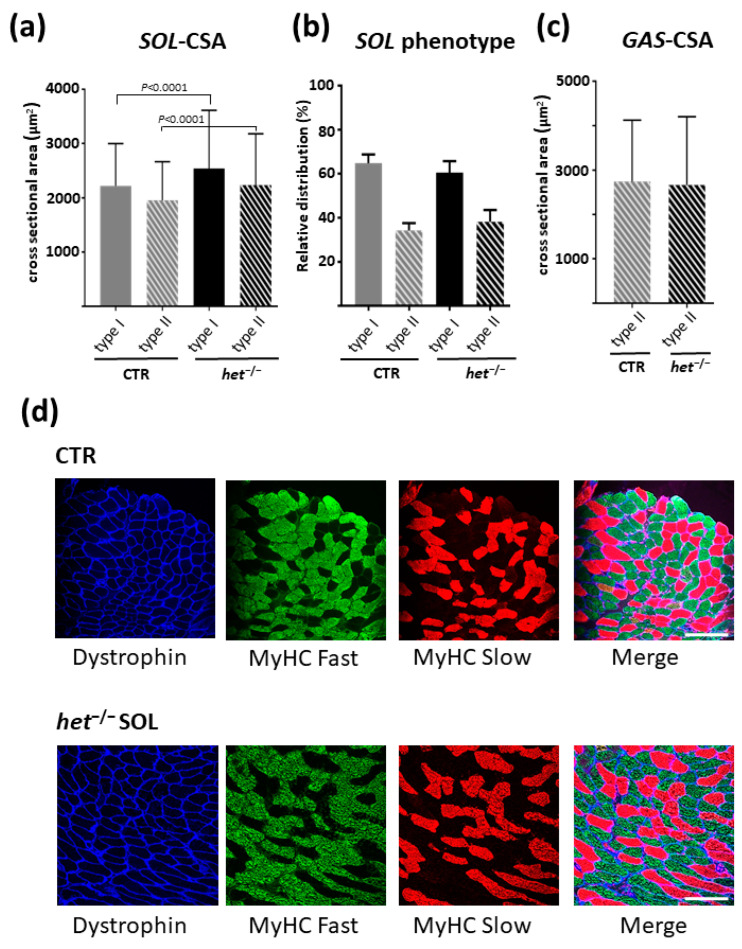
Myofiber cross-sectional area analysis in *het^−/−^* vs. CTR skeletal muscle. (**a**) Graphic represents the CSA of the myofibers in the *SOL* muscle (*het^−/−^ n* = 542 + 775; CTR *n* = 306 + 545 type I and type II fibers, respectively); (**b**) Graphic represents the myofiber phenotype composition of the *SOL* muscle; (**c**) Graphic represents the CSA of the myofibers in the *GAS* muscle (*het^−/−^ n* = 976; CTR *n* = 865; myofibers from six animals/group). The gastrocnemius muscle is comprised of more than 90% type II myofibers; therefore, type I fiber morphological analysis was omitted. (**d**) Representative images from the *SOL* muscle cross sections in the CTR and *het^−/−^* groups, respectively. Dystrophin membrane marker (blue label); type II fast-twitch myofibers (green label); type I slow-twitch myofibers (red label). Magnifican scale bar = 150 μm.

## Data Availability

No new data were created or analyzed in this study. Data sharing is not applicable to this article.

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
