# Peer review of "Increased Homer Activity and NMJ Localization in the Vestibular Lesion het−/− Mouse soleus Muscle"

_ijms, 2024, doi:10.3390/ijms25168577_

Round 1

Reviewer 1 Report

Comments and Suggestions for Authors

The manuscript titled "Increased Homer Cross-Linking Activity and NMJ Subcellular Localization in the Vestibular Lesion het-/- Mouse Soleus Skeletal Muscle" presents a detailed study on the alterations in Homer protein expression and its subcellular localization in the skeletal muscles of het-/- mice, providing significant insights into the impact of vestibular lesions on muscle physiology. The study is well-structured, and the experimental design is solid. However, there are several areas where the manuscript can be improved for greater clarity.

Comments on the Quality of English Language

Author Response

Answers to Reviewer 1 comments

Thank you very much for the critical review and for encouraging us to pursue an accurate revision of the manuscript based on a series of constructive reviewer criticism.

We have taken into considerations all reviewer`s comments and concerns and we have tried to respond point-by-point on most of them.

Overall, we feel confident that we could provide most of the requested answers and explanations.

We are more than confident that this resulted in an improved presentation of our work, making it, hopefully, more suitable for publication in the “International Journal of Molecular Sciences” (IJMS).

In the text of the revised manuscript, all amendments are in red color.

Sincerely yours,

Michele Salanova, PhD.

Reviewer: 1

The manuscript titled "Increased Homer Cross-Linking Activity and NMJ Subcellular Localization in the Vestibular Lesion het-/- Mouse Soleus Skeletal Muscle" presents a detailed study on the alterations in Homer protein expression and its subcellular localization in the skeletal muscles of het-/- mice, providing significant insights into the impact of vestibular lesions on muscle physiology. The study is well-structured, and the experimental design is solid. However, there are several areas where the manuscript can be improved for greater clarity.

Reviewer 1 comment:

The title is informative but somewhat lengthy. I recommend the authors simplify it without losing the essence, for example, "Altered Homer Activity and Localization in Skeletal Muscles of Vestibular Lesion Mice."

Author’s response to concern:

Thank you to the reviewer for the interesting question. Title it has been shortened. Now reads: “Increased Homer Activity and NMJ Localization in the Vestibular Lesion het-/- Mouse soleus Muscle”

Reviewer 1 comment:

The abstract is comprehensive but could be more concise. Focus on the most important findings and their implications. For instance, the details about experimental conditions can be abbreviated. Specify the main findings in terms of percentages or changes to provide a clearer idea.

Author’s response to concern:

Thank you to the reviewer for the interesting question. One entire sentence was replaced in the abstract section lines 27-29: Now a new sentence reads: An increase in Homer cross-linking capacity was present in SOL muscle of het-/- mice as the results of a compensatory mechanisms to the altered vestibule system function. The % increase was add line 34 and line 37.

Reviewer 1 comment:

The introduction could benefit from more recent references to highlight the latest advancements in the field. The connection between vestibular lesions and muscle alterations should be more explicitly stated to underline the study’s significance. I advise the authors to clarify the rationale behind selecting the specific muscles (SOL and GAS) for this study.

Author’s response to concern:

Thank you to the reviewer for the interesting question. Since we have several evidences suggesting that Homer signaling is most important for type-I muscle, the rationale behind selecting the SOL and GAS specific muscles was to compare two totally different types of muscles one which is prevalently “oxidative or slow-type (SOL) with one which is prevalently glycolytic or fast-type muscle (GAS).

Therefore, a new sentence was added to the Introduction section lines 118-119 that reads: Because several lines of evidence showed that Homer signaling is mostly important for slow-twitch skeletal muscles …..

And line 121: “…..mostly represented by oxidative or slow type-I myofibers……”;

line 122: “…..mostly represented by glycolytic or fast type-II myofibers…….”;

line 124: variant.

Two references from the Authors have been taken out.

5 new more recent references have been added to the Introduction, Discussion and Methods section. All are labelled in red in the reference list.

Reviewer 1 comment:

In the results section, the figures should be more self-explanatory with detailed legends. For instance, in Figure 1, I recommend the authors clearly label the bands and include molecular weight markers.

Author’s response to concern:

Thank you to the reviewer for the important question. As suggested by the reviewer, P values were added to the text and the figure legends.

Figure legend 1 and 2 were revised.

A new sentence was added to Figure1 (page 5, lines 179-180) and Figure 2 (page 7, lines 205-206) legends: Data were compared using unpaired Student`s T-test. Considered statistic significant at p< 0,05

Molecular weight markers are included in the new Figure 1.

Reviewer 1 comment:

The authors should provide representative images of the Western blot and immunofluorescence results to enhance reproducibility.

Author’s response to concern:

Thank you to the reviewer for the important question. A new figure 1 was made including  duplicate of soluble and unsoluble fraction of CTR and het-/- from two different mice.

Below there is an extra set of Homer immunofluorescence Merged images of CTR and het-/- SOL muscle for review only.

Reviewer 1 comment:

The statistical analysis section should explicitly state the tests used for each comparison and ensure consistency in reporting p-values. Some results are mentioned in the text but not clearly shown in figures. I advise the authors to ensure all critical data points are visually represented.

Author’s response to concern:

Thank you to the reviewer for the important question. Statistical analysis and the type of test was used are reported in figures legends. P values were also added in the abstract and results section.

Figure 1 lines 168 to 181 now reads:

Figure 1. Homer skeletal muscle Western blot analysis from SOL (a-b) and GAS (c-d) muscle. (a) Representative image of soluble (Cytosol) and unsoluble (Pellet) fractions of CTR and het-/- mice.  Homer dimer (~90 kDa) and Homer multimer (above 150 kDa) immunoreactive bands are present in both soluble and unsoluble fractions of both animal groups. (b) Homer quantification by densitometry analysis in SOL. (c) Representative image of soluble (Cytosol) and unsoluble (Pellet) fractions of CTR and het-/- mice. Homer monomer (~43-45 kDa), dimer (~90 kDa) and multimer (above 150 kDa) immunoreactive bands are detectable mostly in the soluble fraction. No Homer immunoreactive bands are present in the unsoluble fraction of het-/- mice. (d) Homer quantification by densitometry analysis in GAS
Cytosol = soluble fraction; Pellet = unsoluble fraction. Brain was used as positive internal control in panels a and c. Results are from at least three independent experiments. Data were compared using unpaired Student`s T-test. Considered statistic significant at p< 0,05. Alltogether a total of n=6 CTR and n=6 het-/- mice were investigated.

Figure 2, lines 197 to 206 now reads:

Figure 2. Homer immunofluorescence intensity at the NMJ. (a, b) SOL muscle. (a) The graphs show the mean gray scale value obtained in SOL from the Homer signal at the NMJ postsynaptic region. (b) Representative Homer immunofluorescence analysis in CTR and het-/- of NMJ SOL muscle. The number of analyzed NMJs in the SOL: het-/- n=69; CTR n=52. (c, d) GAS muscle. (c) The graphs show the mean gray scale value obtained in GAS from the Homer signal at the NMJ postsynaptic region. (d) Representative Homer immunofluorescence analysis in CTR and het-/- of NMJ GAS muscle. The number of analyzed NMJs in the GAS: het-/- n=45; CTR n=71. In blue= DAPI cell nuclei staining; red = α-Bungarotoxin (BTx)-labelled AChRs; green = Homer proteins. Magnification scale bar = 15µm. Data were compared using unpaired Student`s T-test and/or Mann`-Whitney test. Considered statistic significant at p< 0,05.

Reviewer 1 comment:

The discussion could better integrate the findings with existing literature, emphasizing the novelty and implications of the results. I also recommend stating the limitations of the study and suggesting potential future research directions. The section discussing the potential mechanisms behind increased Homer expression and NMJ reorganization could be more detailed, considering alternative hypotheses and pathways.

Author’s response to concern:

Thank you to the reviewer for the interesting question. A new paragraph as “Study limitation” at the end of the Discussion section was added. Page 13, Lines 405 to 408:

This study is exclusively experimental and has so far only been carried out on muscles of murine models. An equally thorough experimental analysis on muscles of patients with associated balance and locomotion dysfunction is of fundamental importance and necessary in order to be able to draw conclusions pertaining to human pathology.

Reviewer 2 Report

Comments and Suggestions for Authors

The manuscript of Trautmann et al. presents how a recessive variant in the Homer protein, associated with a vestibular disorder can impact the skeletal muscles. In the introduction, the authors explain how the vestibular system could modulate muscle contraction and present evidence to support this hypothesis. Further, the authors focus on the anatomy and biochemistry of the neuromuscular junction and the role of the Homer protein there. In the result section, the Homer monomer and Homer dimer fractions in both cytosolic and non cytosolic fractions have been presented. The authors show increased Ach expression in the NMJ of the SOL and expression of the different isoforms. The results are presented clearly and the figures and graphics are informative. In the discussion, the authors try to link vestibulopathy to muscle pathology and dicuss the most important positive and negativ eresults. The conclusions are concise and based on the results and discussion. The methodology is sound.

It is a well writte and well designed experimental mansucript that deserves publication.

Comment:

I would change the wording "mutation" to "variant"

Author Response

Authors`Response to Reviewer 2 comments

Thank you very much for the critical review and for encouraging us to pursue an accurate revision of the manuscript based on a series of constructive reviewer criticism.

We have taken into considerations all reviewer`s comments and concerns and we have tried to respond point-by-point on most of them.

Overall, we feel confident that we could provide most of the requested answers and explanations.

We are more than confident that this resulted in an improved presentation of our work, making it, hopefully, more suitable for publication in the “International Journal of Molecular Sciences” (IJMS).

In the text of the revised manuscript, all amendments are in red color.

Sincerely yours,

Michele Salanova, PhD.

Reviewer 2

The manuscript of Trautmann et al. presents how a recessive variant in the Homer protein, associated with a vestibular disorder can impact the skeletal muscles. In the introduction, the authors explain how the vestibular system could modulate muscle contraction and present evidence to support this hypothesis. Further, the authors focus on the anatomy and biochemistry of the neuromuscular junction and the role of the Homer protein there. In the result section, the Homer monomer and Homer dimer fractions in both cytosolic and non cytosolic fractions have been presented. The authors show increased Ach expression in the NMJ of the SOL and expression of the different isoforms. The results are presented clearly and the figures and graphics are informative. In the discussion, the authors try to link vestibulopathy to muscle pathology and discuss the most important positive and negative results. The conclusions are concise and based on the results and discussion. The methodology is sound.

Reviewer 2 comment:

It is a well written and well-designed experimental manuscript that deserves publication.

I would change the wording "mutation" to "variant"

Author’s response to concern:

Thank you to the reviewer for the interesting question. The word “mutation” it has been changed to "variant" in the Abstract section line 24 and in the Introduction section page 3 line 124.

Reviewer 3 Report

Comments and Suggestions for Authors

The authors have submitted a research article regarding an evaluation by immunochemical-based image analysis of a possible morphological changes in the het(-/-) mutation mice of soleus skeletal muscle and a possible association of Homer protein linked subcellular signaling system. The authors claimed that skeletal muscle structure and function might be affected by peripheral vestibular system by using het(-/-) mutation mice, illustrating a hypothesis suggesting that the Homer protein signaling system might have a pivotal role in muscle atrophy through vestibular lesions. This issue is of interest, and impact of their results is strong. My overall concern with the article describing the current available data regarding beneficial availability of the evaluation of effects of Homer protein distribution in subcellular portions on muscle structure by using histochemical analysis, offer something substantial that helps advance our understanding of advanced diagnosis and then effective medicinal management available in clinic.

To strengthen authors’ perspectives, the authors are strongly recommended to add a discussion in detail regarding how to evaluate in cases where there is no correlation between the two factors, i.e. image analysis for the muscle structure and the Homer protein expression levels. The opposite outcomes, if known, may influence largely the authors’ perspective.

Author Response

Authors`Response to Reviewer 3 comments

Thank you very much for the critical review and for encouraging us to pursue an accurate revision of the manuscript based on a series of constructive reviewer criticism.

We have taken into considerations all reviewer`s comments and concerns and we have tried to respond point-by-point on most of them.

Overall, we feel confident that we could provide most of the requested answers and explanations.

We are more than confident that this resulted in an improved presentation of our work, making it, hopefully, more suitable for publication in the “International Journal of Molecular Sciences” (IJMS).

In the text of the revised manuscript, all amendments are in red color.

Sincerely yours,

Michele Salanova, PhD.

Reviewer 3

The authors have submitted a research article regarding an evaluation by immunochemical-based image analysis of a possible morphological changes in the het-/- mutation mice of soleus skeletal muscle and a possible association of Homer protein linked subcellular signaling system. The authors claimed that skeletal muscle structure and function might be affected by peripheral vestibular system by using het(-/-) mutation mice, illustrating a hypothesis suggesting that the Homer protein signaling system might have a pivotal role in muscle atrophy through vestibular lesions. This issue is of interest, and impact of their results is strong.

Reviewer 3 comment:

My overall concern with the article describing the current available data regarding beneficial availability of the evaluation of effects of Homer protein distribution in subcellular portions on muscle structure by using histochemical analysis, offer something substantial that helps advance our understanding of advanced diagnosis and then effective medicinal management available in clinic.

Author’s response to concern:

Thank you to the reviewer for the interesting question. However, our data is mainly experimental and it is difficult to draw a conclusion. In any case, based on what is my/our indirect experience, almost all of the patients with balance problems during locomotion are associated with muscle problems and in particular with coordination of leg muscle movement. Nevertheless, there are currently no large case studies or even histochemical analysis of Homer in skeletal muscle of these patients. Therefore, there is no basis for us to draw a conclusion on this.

Reviewer 3 comment:

To strengthen authors’ perspectives, the authors are strongly recommended to add a discussion in detail regarding how to evaluate in cases where there is no correlation between the two factors, i.e. image analysis for the muscle structure and the Homer protein expression levels. The opposite outcomes, if known, may influence largely the authors’ perspective.

Author’s response to concern:

Thank you to the reviewer for the interesting question. A new sentence was added to the Discussion section page 13, lines 381 to 388 that reads:

Different experimental models of disuse-induced muscle atrophy and exercise countermeasure in human and animal models of denervation showed that both Homer expression and subcellular localization at the NMJ are dependent on muscle and nerve activity (34). Therefore, we assume that the increased Homer expression and cross-linking activity in the soleus muscle of het-/- mice, as reported in the present study, are due to an increased muscle and nerve activity caused by an altered vestibular input at the NMJ. This leads in turn to an increase in NMJ signaling which may require an increased recruitment of signaling molecules and thus cross-linking activity of Homer-dependent signaling pathways.

Round 2

Reviewer 1 Report

Comments and Suggestions for Authors

The authors have responded comprehensively to the reviewers' comments.

Comments on the Quality of English Language

The authors have responded comprehensively to the reviewers' comments.

Reviewer 3 Report

Comments and Suggestions for Authors

The authors have done a good job responding to reviewer comments and concerns in their revision. I believe the manuscript is significantly improved as a result. Now I recommend that this revised version of the manuscript can be accepted for publication in the journal IJMS.